# Docking Analysis of Some Bioactive Compounds from Traditional Plants against SARS-CoV-2 Target Proteins

**DOI:** 10.3390/molecules27092662

**Published:** 2022-04-20

**Authors:** Nourhan M. Abd El-Aziz, Ibrahim Khalifa, Amira M. G. Darwish, Ahmed N. Badr, Huda Aljumayi, El-Sayed Hafez, Mohamed G. Shehata

**Affiliations:** 1Department of Food Technology, Arid Lands Cultivation Research Institute (ALCRI), City of Scientific Research and Technological Applications (SRTA-City), Alexandria 21934, Egypt; amiragdarwish@yahoo.com (A.M.G.D.); gamalsng@gmail.com (M.G.S.); 2Food Technology Department, Faculty of Agriculture, Benha University, Moshtohor, Benha 13736, Egypt; ibrahiem.khalifa@fagr.bu.edu.eg; 3Department of Food Toxicology and Contaminants, National Research Centre, Dokki, Cairo 12622, Egypt; noohbadr@gmail.com; 4Department of Food Science and Nutrition, College of Sciences, Taif University, P.O. Box 11099, Taif 21944, Saudi Arabia; huda.a@tu.edu.sa; 5Department of Plant Protection and Biomolecular Diagnosis, Arid Lands Cultivation Research Institute (ALCRI), City of Scientific Research and Technological Applications (SRTA-City), Alexandria 21934, Egypt; elsayed_hafez@yahoo.com; 6Food Research Section, R&D Division, Abu Dhabi Agriculture and Food Safety Authority (ADAFSA), Abu Dhabi P.O. Box 52150, United Arab Emirates

**Keywords:** traditional plants, SARS-CoV-2, bioactive compounds, molecular docking, pharmacokinetic

## Abstract

COVID-19 is still a global pandemic that has not been stopped. Many traditional medicines have been demonstrated to be incredibly helpful for treating COVID-19 patients while fighting the disease worldwide. We introduced 10 bioactive compounds derived from traditional medicinal plants and assessed their potential for inhibiting viral spike protein (S-protein), Papain-like protease (PLpro), and RNA dependent RNA polymerase (RdRp) using molecular docking protocols where we simulate the inhibitors bound to target proteins in various poses and at different known binding sites using Autodock version 4.0 and Chimera 1.8.1 software. Results found that the chicoric acid, quinine, and withaferin A ligand strongly inhibited CoV-2 S -protein with a binding energy of −8.63, −7.85, and −7.85 kcal/mol, respectively. Our modeling work also suggested that curcumin, quinine, and demothoxycurcumin exhibited high binding affinity toward RdRp with a binding energy of −7.80, −7.80, and −7.64 kcal/mol, respectively. The other ligands, namely chicoric acid, demothoxycurcumin, and curcumin express high binding energy than the other tested ligands docked to PLpro with −7.62, −6.81, and −6.70 kcal/mol, respectively. Prediction of drug-likeness properties revealed that all tested ligands have no violations to Lipinski’s Rule of Five except cepharanthine, chicoric acid, and theaflavin. Regarding the pharmacokinetic behavior, all ligand predicted to have high GI-absorption except chicoric acid and theaflavin. At the same way chicoric acid, withaferin A, and withanolide D predicted to be substrate for multidrug resistance protein (P-gp substrate). Caffeic acid, cepharanthine, chicoric acid, withaferin A, and withanolide D also have no inhibitory effect on any cytochrome P450 enzymes. Promisingly, chicoric acid, quinine, curcumin, and demothoxycurcumin exhibited high binding affinity on SARS-CoV-2 target proteins and expressed good drug-likeness and pharmacokinetic properties. Further research is required to investigate the potential uses of these compounds in the treatment of SARS-CoV-2.

## 1. Introduction

Coronavirus disease 2019 (COVID-19), the highly infectious viral illness caused by severe acute respiratory syndrome coronavirus 2 (SARS-CoV-2), has had a disastrous effect on the world’s demographics, resulting in more than 4.5 million deaths worldwide. SARS-CoV-2 is a positive-sense RNA (30 kb) virus. Two types of proteins characterize the human coronaviruses HCoVs, structural proteins (Spike (S), Nucleocapsid (N), Matrix (M), and Envelope (E)), and non-structural proteins including the RdRp [1,2]. The membrane and envelope proteins are associated with virus assembly, while the spike protein plays the main role in facilitating virus entry via mediating its interaction with the transmembrane surface receptor on the host cells [3].

This virus contains S- protein, PLpro, and RdRp involved in viral entry, replication, and immune response evasion. Drugs targeting these proteins therefore have great potential for inhibiting the virus. The spike protein directly interacts with the peptidase domain (PD) of the angiotensin-converting enzyme 2 (ACE2) receptor [4], which technically marks the virus entry inside the cells [5]. ACE2 interaction with spike protein could aid in producing antivirals or a vaccine that can block CoV infection by targeting ACE2 [6]. The binding to the SARSCoV-2 spike protein at the same place where the original ACE2 PD domain interacts, which shows that 23-amino acids sequence independently has the potential to inhibit the interaction of the SARS-CoV-2 S- protein and ACE2 complex. Hence, the viral entrance. Interacting residues at the interface of the SARS-CoV-2 and ACE2 PD domain are ARG403 LYS417 GLY446 TYR449 TYR453 LEU455 PHEF456 TYR473 ALA475 GLY476 GLY485 PHE486 ASN487 TYR489 GLN493 TYR495 GLY496 GLNQ498 THR500 ASN501 GLY502 VAL503 TYR505 in SARS-CoV-2 S- protein [7].

Another viral protein, PLpro, which is vital for viral replication [8], is responsible for the proteolytic processing of the product of open reading frame 1a (ORF1a) in the replicase gene of CoV2, a large viral polyprotein containing non-structural proteins which form the replicase complex [9]. The peptide bond cleavage in the active site is catalyzed by a conserved catalytic triad comprised of residues CYS111, HIS272, and ASP286 [10]. In addition, PLpro possesses deubiquitinating and deISGylating capabilities [11] which interfere with critical signaling pathways leading to the expression of type I interferons, resulting in antagonistic effect on host innate immune response [12]. Therefore, inhibition of PLpro activity can halt viral replication and disrupt its role in host immune response evasion, making it an excellent anti-viral drug target. RdRp is also a vital enzyme for the life cycle of RNA viruses. It has been targeted in various viral infections (Hepatitis C virus (HCV), Zika virus (ZIKV), and human coronaviruses (HCoV) [13]. SARS-CoV-2 polymerase residues ARG 553, ARG 555, LYS 545, and ASN 691 interaction with some bioactive compounds play an important role in virus replication inhibition [14]. Thus, many research working in computer-aided drug design in generating new small molecules that might interact with SARS-CoV-2 S -protein, RdRp, and 3CL or Mpro protease proteins to combat the COVID-19 disease.

In the U.S., the Food and Drug Administration issued an emergency use authorization (EUA) for Pfizer’s Paxlovid (nirmatrelvir tablets and ritonavir tablets, co-packaged for oral use) for the treatment of mild-to-moderate coronavirus disease (COVID-19) in adults and pediatric patients (12 years of age and older weighing at least 40 kg or about 88 pounds) with positive results of direct SARS-CoV-2 testing, and who are at high risk for progression to severe COVID-19, including hospitalization or death. Paxlovid consists of nirmatrelvir, which inhibits a SARS-CoV-2 protein to stop the virus from replicating, and ritonavir, which slows down nirmatrelvir’s breakdown to help it remain in the body for a longer period at higher concentrations. Paxlovid is available by prescription only and should be initiated as soon as possible after diagnosis of COVID-19 and within five days of symptom onset [15]. Another antiviral drug use for the same purpose is Lagevrio (molnupiravir). Molnupiravir is a nucleoside analogue that inhibits SARS-CoV-2 replication by viral mutagenesis. The U.S. FDA has issued an EUA to make Molnupiravir available during the COVID-19 pandemic (Molnupiravir is not FDA-approved for any uses, including use as treatment for COVID-19. This medicine is still being studied, Molnupiravir may cause serious side effects [16].

The curative effects of traditional herbal formula sometimes are not necessarily by directly inhibiting or killing the virus, but through the integration of various aspects such as relieving cytokines storm syndrome, protecting human tissues and organs, relieving immunological injury, and enhancing the body’s ability [17].

Ayurveda, known as “The Science of Life,” an ancient traditional medicinal system that originated and is practiced in India, has been utilized for reducing SARS-CoV-2 infection and treating COVID-19-associated patients [18,19,20]. It describes many medicinal plants and herbs possessing a broad range of therapeutic usefulness in curing various kinds of ailments, diseases, and disorders, such as *Allium sativum* (Garlic), *W. somnifera* known commonly as ashwagandha, *Zingiber officinale* Roscoe (Ginger), *Tinospora cordifolia* (Giloy), *Ocimum sanctum* (Tulsi), *Curcuma longa* (Turmeric, Haldi), *Glycyrrhiza glabra* (Licorice, mulethi), and others [20,21,22].

In this study we aim to investigate docking characteristics of 10 bioactive compounds identified from traditional medicines (caffeic and chicoric acids from *Echinacea purpurea*; xanthorrhizol from *Java turmeric*; curcumin and demothoxycurcumin from *Curcuma longa* L.; Quinine from *Chinchona bark*; Theaflavin from *Black tea*; withaferin A and withanolide D from *Withania somnifera* L.; cepharanthine from *Stephania cepharantha*) on S-protein, RdRp and SARS-CoV-2 PLpro to explore the potential mode of the inhibitory activity toward SARS-CoV-2. The molecules that showed better hydrogen bonding and binding energy were selected further for drug likeness and pharmacokinetics prediction.

## 2. Results and Discussion

Herbal medicines and medicinal plant-based natural compounds provide a rich resource for novel antiviral drug development. Some natural medicines have been shown to possess antiviral activities against various virus strains including coronavirus, herpes simplex virus [23,24,25], influenza virus [25], human immunodeficiency virus [26], hepatitis B and C viruses [27], SARS, and MERS [28,29]. These compounds antiviral action mechanisms caused by influencing of the viral life cycle, such as viral entry, replication, assembly, and release, as well as virus-host-specific interactions. In this study we analyze the binding mode of 10 ligands Figure 1 that previously identified in some traditional plants Table 1 using computational docking analysis.

All tested ligands showed hydrogen interaction to papain-like protease (6W9C) except cepharanthine and theaflavin. The ligand docking on CYS111, HIS272, and ASP286 catalytic residues (Figure 2A) ranked by binding energy ΔG in kcal/mol were as follows: chicoric acid > demothoxycurcumin > curcumin > withanolide D > xanthorrhizol > withaferin A > caffeic acid > quinine. It is known that the high binding affinity of the drug compounds depends on the type and amount of bonds occurring with the target protein [53]. The binding energies were −7.62, −6.81, −6.70, −6.58, −6.16, −6.13, −5.70, and −5.67 kcal/mol, respectively (Table 2; Figure 3).

The first reported bioactive effect of chicoric acid is its ability to inhibit infection with human immunodeficiency virus 1 (HIV-1) [54]. Several studies reported that chicoric acid inhibits infection with HIV-1 by deactivating the HIV-1 integrase. HIV-1 integrase is a multidomain enzyme required for integration of viral DNA into the host genome, a critical step in viral replication [55]. Inhibition of HIV-1 integration by chicoric acid results in stopping virus replication, leading to increased T-lymphoblastoid cell viability [56]. Chicoric acid found in many Egyptian source as *Chicory* and *Echinacea purpurea* (Table 1). Despite that cepharanthine (CEP) showed no binding behavior toward papain-like protease, it has unique anti-inflammatory, antioxidative, immunomodulating, antiparasitic, and antiviral properties. It can suppress nuclear factor-kappa B (NF-κB) activation, lipid peroxidation, nitric oxide (NO) production, cytokine production, and expression of cyclooxygenase; all of which are crucial to viral replication and inflammatory response [33]. Against SARS-CoV-2 and homologous viruses, CEP predominantly inhibits viral entry and replication at low doses [57].

Moreover, all tested ligands showed H-interaction to spike protein (6M17). The ligand docking on GLY 502, TYR 489 and TYR 505 (Figure 2B) ranked by binding energy in kcal/mol were as follows: chicoric acid > quinine = withaferin A > withanolide D > theaflavin > demothoxycurcumin > curcumin > xanthorrhizol > cepharanthine > caffeic acid. The binding energies were −8.63, −7.85, −7.85, −7.78, −7.43, −7.23, −6.34, −6.00, −5.26 and −5.06 kcal/mol respectively (Table 2; Figure 4). Cepharanthine (CEP) is a natural alkaloid, which has been widely used to treat many of the acute and chronic diseases. It is usually used in Japan in the form of *Stephania cepharantha* and *Stephania rotunda* (Table 1) for its anti-inflammatory, antiparasitic, anti-oxidative, antiviral, and anti-HIV activities [32,33,34,35]. In 2021 a Japanese research team reported the anti-COVID-19 activity of the combining cepharanthine/nelfinavir providing synergistic antiviral effects that CEP and nelfinavir inhibit SARS-CoV-2 entry and RNA replication, respectively [58]. In 2021, a Japanese research team reported that the combination of cepharanthine and nelfinavir inhibited SARS-CoV-2 entrance and RNA replication, offering synergistic antiviral effects [58].

This previous combination based on in silico docking simulation that confirms that CEP molecule can bind to SARS-CoV-2 S-protein and interfere with the spike engagement to its receptor, angiotensin-converting enzyme 2 (ACE2) [59,60]. Our results strongly supported this mechanism of CEP toward SARS-CoV-2 S-protein. However, chicoric acid, quinine, withaferin A, withanolide D, theaflavin, demothoxycurcumin, curcumin, and xanthorrhizol expressed higher binding energy than cepharanthine in case SARS-CoV-2 S-protein binding. From all of this we supposed that chicoric acid and quinine may play the same role in this combination (cepharanthine/nelfinavir) beside their acting mechanism toward SARS-CoV-2, PLpro and RdRp that may be promising to limit SARS-CoV-2 proliferation than CEP.

Regarding to RdRp (6M71), all tested ligands showed H-interaction except Cepharanthine. The ligand docking on ARG 553, ARG 555, LYS 545, and ASN 691 (Figure 2C) ranked by binding energy ΔG in kcal/mol were as follows: curcumin = quinine > demothoxycurcumin > chicoric acid > caffeic acid > withaferin A > withanolide D > xanthorrhizol > theaflavin. The binding energies were −7.80, −7.80, −7.64, −7.50, −6.75, −6.27, −6.10, −5.33, and −2.55 kcal/mol, respectively (Table 2; Figure 5).

Curcumin (CC) and its analogues are the main phytonutrients of turmeric (*Curcuma longa* L.) and other *Curcuma* spp., which are widely used around the world as culinary spices, traditional medicine as well as a popular dietary supplement ingredient due to its wide range of health benefits including anti-inflammation [61], anti-cancer [62], cardiovascular regulation [63], respiratory [64], and immune system benefits [65]. In addition, the suppression of multiple cytokines by curcumin suggested that it may be a useful approach in treating Ebola patients against cytokine storm [66]. As severe cases of COVID-19 are often associated with cytokine release syndrome, the use of anti-inflammatory molecules may reduce the proinflammatory cytokines involved. Quinine, is an extract of the bark of the Chinchona tree (native to the Andes of South America), that was used to treat feverish infections, particularly malaria, for hundreds of years almost worldwide [67]. *Curcuma longa* L. and *Chinchona bark* are rich source of curcumin and quinine respectively (Table 1).

Table 3 presents the drug likeness and pharmacokinetic properties of the tested ligands. Lipinski’s Rule of Five is generally used as an indicator of the drug likeness and pharmacological activities. In humans, this would make them orally active medications [68]. All the tested compounds are in the molecular weight range of 180.16 to 474.37 Da (<500 Da), except for cepharanthine and theaflavin are 606.71 and 564.49 Da. It is well known that drug molecules typically have low molecular weight (<500 kDa) are transported, diffused, and absorbed easily compared with large molecules [69]. All the tested compounds also have fewer than 15 rotatable bonds, and all have less than 5 hydrogen bond donors (NH and OH), except for quinine and theaflavin. In addition, the numbers of hydrogen bond acceptors (O and N atoms) predicted in all compounds are less than 10, except for chicoric acid and theaflavin (Table 3). At the same time, permeability (logP) of these ligands has also been investigated, and it was found that these ligands exhibited logP values of less than 5 except cepharanthine. Moreover, the topological polar surface area (TPSA) values of all ligands are less than 140 Å, except for chicoric acid and theaflavin. TPSA is closely related to the H-bonding ability of a compound [70]. The TPSA and the logP values are the two essential characteristics in the analysis of the bioavailability of drug molecules and permeability toward bio-membranes [71]. Compounds with 10 rotatable bonds and TPSA of ≤140 Å can be predicted to have good bioavailability [72]. As a result, all of the ligands examined have good bioavailability, with the exception of chicoric acid and theaflavin, which have TPSA values of 208.12 and 217.60, respectively. Pharmacokinetic prediction provides information to evaluate the time course of drugs and their effects on the body to design an appropriate drug regimen for any patient [73].

Our results of pharmacokinetic prediction (Table 3) showed that all tested ligands demonstrate a high absorption rate in the GI tract, except for chicoric acid and theaflavin. Moreover, none of the tested ligands were predicted to be able to pass through the blood–brain barrier, except for quinine and xanthorrhizol. Additionally, chicoric acid, withaferin A and withanolide D are the only compounds predicted to act as a substrate for P-gp, which decrease their further clinical application [74]. The predicted pharmacokinetic behavior indicates that only caffeic acid, cepharanthine, chicoric acid, withaferin A, and withanolide D would have no inhibitory effect on any cytochrome P450 enzymes (CYP1A2, CYP2C19, CYP2D6, and CYP3A4). Furthermore, CYP2C9 activity is only known to be affected by curcumin, demothoxycurcumin, theaflavin and xanthorrhizol. Cytochrome P450 enzymes are essential for the metabolism of many chemicals. Although this class contains more than 50 enzymes, six of them metabolize 90% of compounds. Cytochrome P450 enzyme inhibition causes unwanted adverse effects or therapeutic failures [75]. The proposed mechanism of SARS-CoV-2 inhibition by tested compounds is summarized in Figure 6; chicoric acid, quinine and withaferin A, interacting with S protein, inhibits viral ability to attach to human ACE2. Additionally, chicoric acid, demothoxycurcumin, and curcumin interacting with PL pro, inhibits viral replication, moreover, curcumin, demothoxycurcumin, and quinine interacting with RdRp, inhibits viral replication. Furthermore, interferon (IFN) production is triggered by virus infections that cause accumulation of RNA species in the cytoplasm, initiate the oligomerization of an essential mitochondrial antiviral signaling protein, MAVS, that serves as a scaffold for the activation of TNF receptor-associated factor (TRAF) family proteins, TRAF2, TRAF3, TRAF5, and TRAF6. TRAF proteins catalyze the assembly of K63-linked ubiquitin chains that are required for the activation of serine kinases, IKKa, IKKb, IKKc, IKKe, and TBK1. Kinase activation triggers the phosphorylation and nuclear import of IRF3, driving the production of primary antiviral effectors including IFNs [76]. On the other hand, PLpro of SARS-CoV-2 interfere with innate immune response by directly cleaving IRF3, inhibits IFNβ production. In general, in silico studies are a great starting point when looking for new therapeutic targets. In the event of an unpredictable pandemic, such as the recent COVID-19 epidemic, in silico research are critical. Drug targets cannot be discovered in a short period of time using existing methods. To reach bulk targets, it is unavoidable to have extremely rapid procedures. In silico approaches can be used to speed up the search for drug targets in natural medicinal plants. This article’s results warrant further evaluation of the potential anti-SARS-CoV-2 activity of these ligands in vitro.

Steps from one to six describe the replication cycle of SARS-CoV-2; attachment (1), viral entry (2), translation of viral proteins (3), viral replication (4), assembly (5), and release (6). In red: chicoric acid, quinine, and withaferin A, interacting with S protein, inhibiting its viral ability to attach to human ACE2 (1). Additionally, chicoric acid, demothoxycurcumin, and curcumin interacting with PL pro inhibits viral replication (2), moreover, curcumin, demothoxycurcumin, and quinine interacting with RdRp inhibits viral replication (3). Interferon (IFN) production is triggered by virus infections that cause accumulation of RNA species in the cytoplasm, initiating the oligomerization of an essential mitochondrial antiviral signaling protein, MAVS, which serves as a scaffold for the activation of TNF receptor-associated factor (TRAF) family proteins, TRAF2, TRAF3, TRAF5, and TRAF6. TRAF proteins catalyze the assembly of K63-linked ubiquitin chains that are required for the activation of serine kinases, IKKa, IKKb, IKKc, IKKe, and TBK1. Kinase activation triggers the phosphorylation and nuclear import of IRF3, driving the production of primary antiviral effectors including IFNs [74]. In blue: PLpro of SARS-CoV-2 interfere with innate immune response by directly cleaving IRF3, inhibits IFNβ production.

## 3. Experimental Section

### 3.1. Docked SARS-CoV-2 Protein Structures

The structure of the SARS-CoV-2 papain-like protease (6W9C), RNA-dependent RNA polymerase (6M71), and spike protein (6M17) used as a target for ligands binding was downloaded from RCSB website [77]. PDB (Protein Data Bank) has enabled breakthroughs in research, such as this study, and education worldwide [78].

### 3.2. Ligands and Drug Scan

The three-dimensional (3D) structures of all tested ligands were drawn in ACD/ChemSketch and then docked into the rigid binding pocket of 6W9C, 6M71, and 6M17 of SARS CoV-2. The compounds used in the present study were caffeic acid, cepharanthine, chicoric acid, curcumin, demothoxycurcumin, quinine, theaflavin, withaferin a, withanolide d and xanthorrhizol. The drug likeness and pharmacokinetic properties were calculated using the SWISSADME prediction website (http://www.swissadme.ch/) (accessed on 15 December 2021) [68,79]. Figure 1 shows the structure of the 10 docked ligands we used herein.

### 3.3. Determination of SARS-CoV-2 Proteins Binding Hits

Table 4 contains the amino acids binding hits of the SARS-CoV-2 polymerase as described by Afonine et al. [80], papain-like protease (6W9C) as described by Baez-Santos et al., 2015 and spike protein (6M17) as described by Baig et al. [7].

### 3.4. Molecular Docking

Ligand optimization was performed using Open Babel, converting ligands from mol into the PDB format. Autodock version 4.0 was used for protein optimization through the removal of water and other atoms and then addition of a polar hydrogen group. Ligand tethering of the protein was performed by regulating the genetic algorithm (GA) parameters using 10 runs of the GA criteria. Docking analyses and determination of hydrogen bonds (H-bonds) were conducted using Chimera 1.8.1 [81].

## 4. Conclusions

Antiviral treatments targeting the coronavirus disease 2019 are urgently required. We screened a panel of ligands from traditional plant sources already used in medicinal fields and some of them described for severe acute respiratory syndrome coronavirus 2 (SARS-CoV-2). Chicoric acid, quinine, curcumin, and demothoxycurcumin exhibited high binding affinity on SARS-CoV-2 S- protein, PLpro, and RdRp. Binding of these proteins interfere with the viral entry, replication, and immune response evasion. Therefore, these compounds may have a great potential for inhibiting the virus. Our work provides data about the ligand mechanism toward SARS-CoV-2 target proteins and allows the comparison between them based on scientific information. Further in vitro cell-based investigations will be needed for chicoric acid, quinine, curcumin, and demothoxycurcumin compounds to confirm their previous results and determine their antiviral mechanism at cell level.

## Figures and Tables

**Figure 1 molecules-27-02662-f001:**
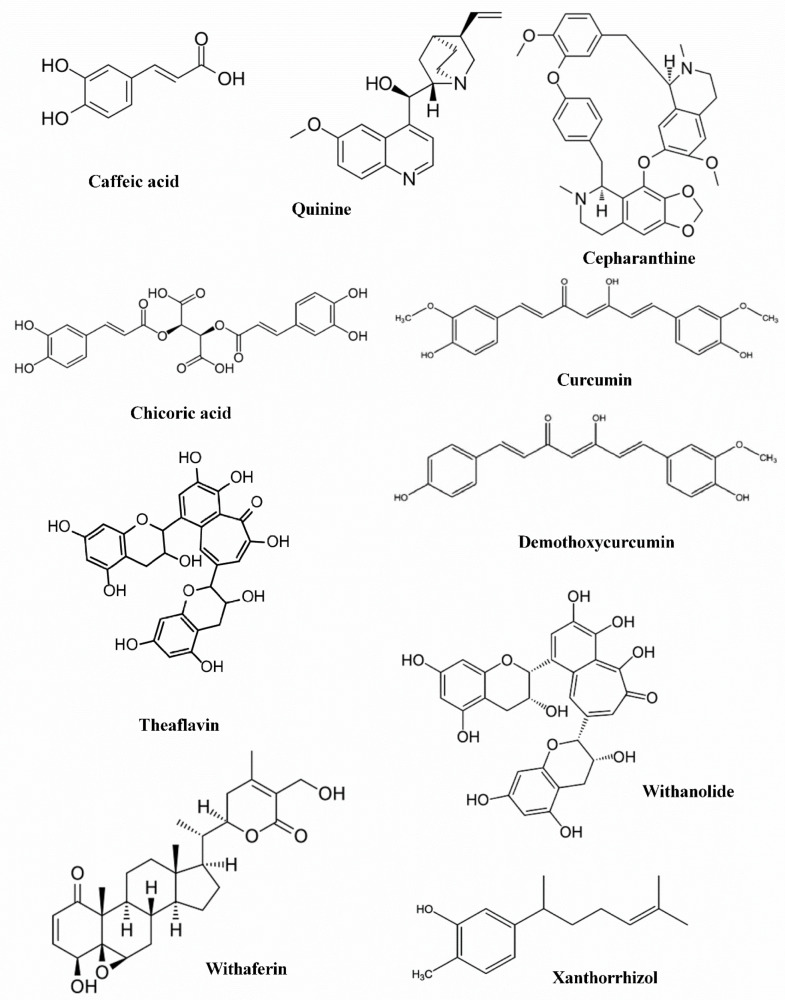
Structures of docking ligands from ZINC drug database.

**Figure 2 molecules-27-02662-f002:**
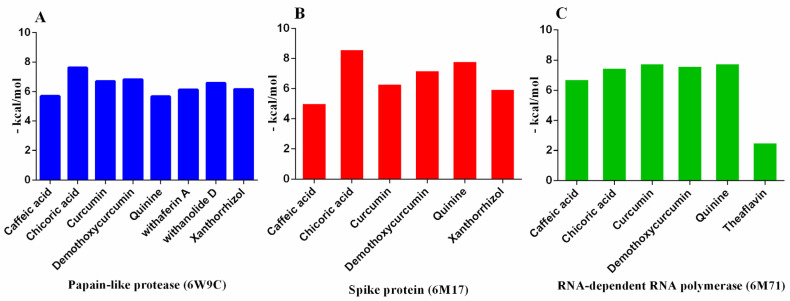
Histogram showing molecular docking results between (**A**) 6W9C, (**B**) 6M17, and (**C**) 6M71 and several drug candidates compounds (the binding energy value ΔG is shown in minus kcal/mol).

**Figure 3 molecules-27-02662-f003:**
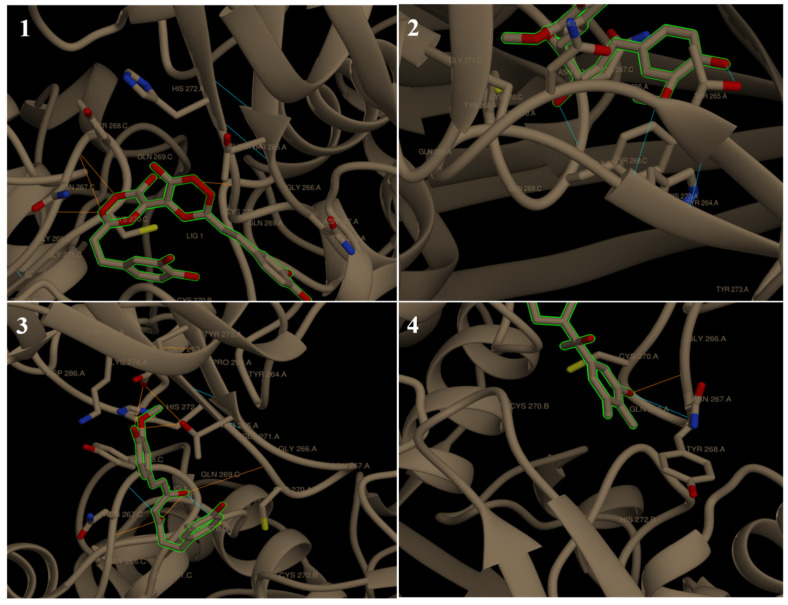
Chimera visualization of 6W9C docking with chicoric acid (**1**), curcumin (**2**), demothoxycurcumin, and (**3**) withanolide D (**4**). The yellow dots show H-bonds.

**Figure 4 molecules-27-02662-f004:**
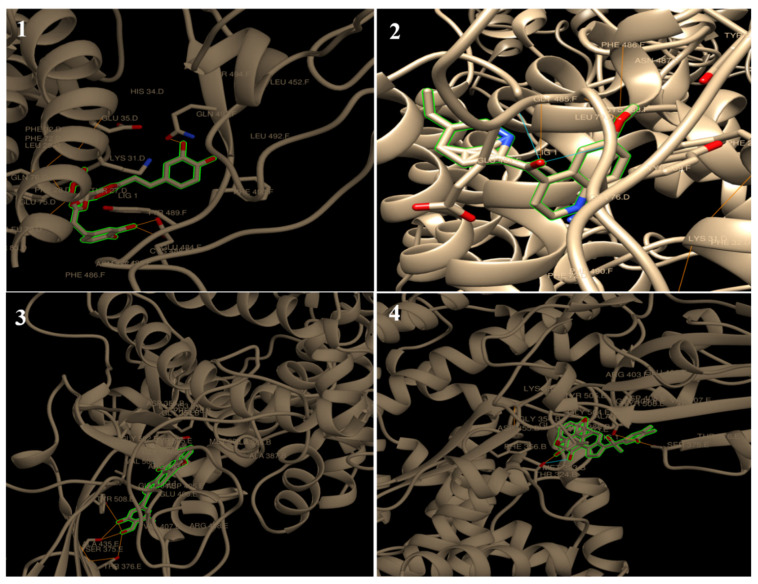
Chimera visualization of 6M17 docking with chicoric acid (**1**), quinine (**2**), withaferin A (**3**) and withanolide D (**4**). The yellow dots show H-bonds.

**Figure 5 molecules-27-02662-f005:**
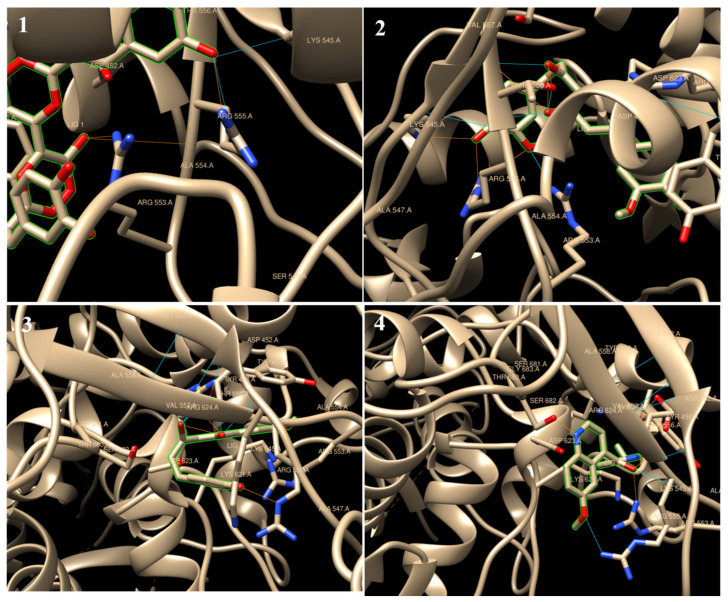
Chimera visualization of 6M71 docking with chicoric acid (**1**), curcumin (**2**), demothoxycurcumin (**3**) and quinine (**4**). The yellow dots show H-bonds.

**Figure 6 molecules-27-02662-f006:**
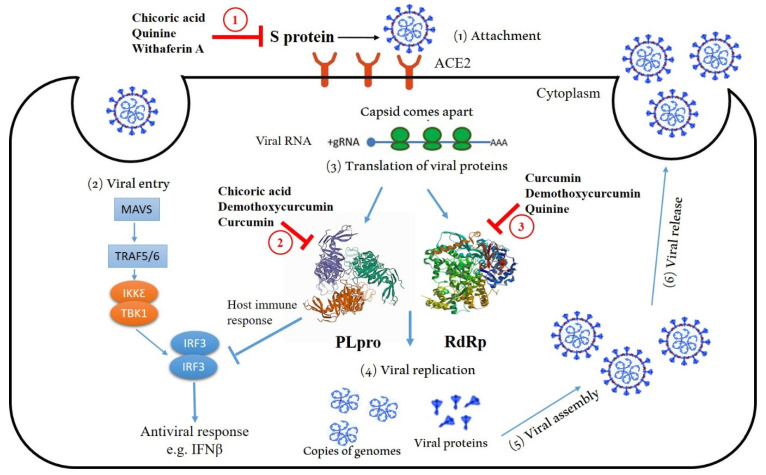
The proposed mechanism of SARS-CoV-2 inhibition by tested compounds.

**Table 1 molecules-27-02662-t001:** Traditional medicinal source for bioactive compounds docking ligands toward COVID-19.

Ligands	Traditional Medicine Source	Region	Reported Biological Activity	References
1	*Citrullus colocynthis* (fruit coat), *C. colocynthis* (Seeds), *Paraguariensis* *Mesona chinensis* *Echinacea purpurea*	Egypt, Poland, Argentina, China	Antiviral activities Antioxidant activity	[30] [31]
2	*Stephania cepharantha* *Stephania rotunda*	Japan	Anti-inflammatory activity, Antiparasitic activities, Anti-oxidative properties, Antiviral activities and Anti-HIV activity	[32,33,34,35]
3	*Sickle seagrass, starflower, sweet basil, African basil, Chicory and Echinacea purpurea*	Egypt, China, India, and North America	Antivirus, anti- inflammation, glucose and lipid homeostasis, Neuroprotection, antioxidation effects. Antimicrobial activity and Antioxidant activity	[36,37,38]
4	*Curcuma longa* L.	Southern Asia, China, India, Indonesia, Indochina	Anti-inflammatory activity, Antioxidant activity, Anti-bacteria activity Chemopreventive and Chemotherapeutic activity Anti-HIV activity, and Nematocidal activities	[39,40,41]
5	*Curcuma longa* L.	Southern Asia, China, India, Indonesia, Indochina,	Antimicrobial activity, Anti-inflammatory activity, Antioxidant activity, Platelet aggregation inhibitory activity, Antiallergy activity, Anticancer activity	[42,43,44]
6	*Chinchona bark*	Peru, Bolivia, Colombia, Ecuador, India, and Sri Lanka.	Antimalarial activity, Antioxidant activity, Anti-cancer agent, Anti-inflammatory, Antiparasitic activity and Antimicrobial property	[45]
7	*Black tea*	Asia and Europe	Antioxidant activity and Anti-cancer activity	[46,47]
8	*Withania somnifera* L.	Sri Lanka	Anti-cancer activity and Anti-COVID activity	[48]
9	*Withania somnifera* L. *Datura metel* L. leaves	India	Anti-inflammatory and Antioxidant activity	[49,50]
10	Java turmeric *Curcuma* species (*C. zedoaria*, *C*. *xanthorrhiza*, *C. aeruginosa* and *C. mangga*)	India and Southeast Asia	Anti-inflammatory, Antioxidant, and Anti-cancer activities	[51,52]

1: Caffeic acid; 2: Cepharanthine; 3: Chicoric acid; 4: Curcumin; 5: Demothoxycurcumin; 6: Quinine; 7: Theaflavin; 8: withaferin A; 9: withanolide D; 10: Xanthorrhizol.

**Table 2 molecules-27-02662-t002:** Molecular docking analysis of several compounds against papain-like protease (6W9C), RNA-dependent RNA polymerase (6M71), and spike protein (6M17) of SARS CoV-2.

Ligand Name	1	2	3	4	5	6	7	8	9	10
Molecular formula	C_9_H_8_O_4_	C_37_H_38_N_2_O_6_	C_22_H_18_O_12_	C_21_H_20_O_6_	C_20_H_18_O_5_	C_20_H_24_N_2_O_2_	C_29_H_24_O_12_	C28H38O6	C28H38O6	C_15_H_22_O
Classification	Phenolics	Alkaloid	Phenylpropanoid	Phenolics	Phenolics	Alkaloid	Phenolics	Phenolics	Phenolics	Sesquiterpenoid
6W9C										
Binding energy ΔG	−5.70	-	−7.62	−6.70	−6.81	−5.67	-	−6.13	−6.58	−6.16
No. of H bonding	6	-	4	3	6	2	-	1	2	2
Binding sites	CYS 270, GLN 269, TYR 268, ASN 267 and THR 265	-	CYS 270, TYR 268, ASN 267 and ASP 286	THR 265 and TYR 268	CYS 270, TYR 268, ASN 267, THR 265 and GLU 263	ASP 286 and LEU 290	-	TYR 264	ASN 267 and TYR 268	ASN 267 and TYR 268
6M71										
Binding energy ΔG	−6.75	-	−7.50	−7.80	−7.64	−7.80	−2.55	−6.27	−6.10	−5.33
No. of H bonding	6	-	4	11	8	4	7	4	7	2
Binding sites	THR 680, ARG 555, ARG 553, THR 556, and SER 682	-	ALA 554, ARG 553 and ARG 555	ARG 555, ARG 553, THR 556, and LYS 545	ARG 555, and THR 556,	ARG 555, THR 556, SER 682 and ALA 554	ARG 553, THR 556, and SER 682	THR 556, and SER 682	THR 556, SER 682 and ASP 623	TYR 455 and ALA 554
6M17										
Binding energy ΔG	−5.06	−5.26	−8.63	−6.34	−7.23	−7.85	−7.43	−7.85	−7.78	−6.00
No. of H bonding	2	1	4	4	1	4	7	12	7	3
Binding sites	ASN 487	GLN 76	GLU 484, CYS 488, GLN 493 and GLN 76	ASP 355, THR 500 and THR 324	ASN 487	CYS 488, GLY 484 and ASN 487	THR 324, GLY 404, ARG 408 and THR 508	THR 324, SER 375, TYR 376 and TYR 508	THR 324, SER 375, and TYR 508	THR 324 and VAL 503

1: Caffeic acid; 2: Cepharanthine; 3: Chicoric acid; 4: Curcumin; 5: Demothoxycurcumin; 6: Quinine; 7: Theaflavin; 8: withaferin A; 9: withanolide D; 10: Xanthorrhizol.

**Table 3 molecules-27-02662-t003:** Predicted drug likeness and pharmacokinetics of SARS-CoV-2 potential inhibitors.

Ligands Name	1	2	3	4	5	6	7	8	9	10
**Lipinski’s Rule of Five**										
Molecular weight (<500 Da)	180.16	606.71	474.37	368.38	338.35	324.42	564.49	470.60	470.60	218.33
LogP (<5)	0.93	5.35	1.01	3.03	3.00	2.81	1.31	2.29	2.36	4.34
No. rotatable bonds (<15)	2	2	11	8	7	4	2	3	2	4
No. H-Bond donors (5)	3	0	6	2	2	1	9	4	4	1
No. H-bond acceptors (<10)	4	8	12	6	5	4	12	6	6	1
TPSA Å	77.76	61.86	208.12	93.06	83.83	45.59	217.60	115.06	115.06	20.23
Violations	0	0	0	0	0	0	3	0	0	0
**Pharmacokinetics**										
GI absorption	High	High	Low	High	High	High	Low	High	High	High
BBB	No	No	No	No	No	Yes	No	No	No	Yes
P-gp substrate	No	No	Yes	No	No	No	No	Yes	Yes	No
CYP1A2 inhibitor	No	No	No	No	Yes	No	No	No	No	No
CYP2C19 inhibitor	No	No	No	No	No	No	No	No	No	No
CYP2C9 inhibitor	No	No	No	Yes	Yes	No	Yes	No	No	Yes
CYP2D6 inhibitor	No	No	No	No	No	Yes	No	No	No	Yes
CYP3A4 inhibitor	No	No	No	Yes	Yes	No	Yes	No	No	No

1: Caffeic acid; 2: Cepharanthine; 3: Chicoric acid; 4: Curcumin; 5: Demothoxycurcumin; 6: Quinine; 7: Theaflavin; 8: withaferin A; 9: withanolide D; 10: Xanthorrhizol.

**Table 4 molecules-27-02662-t004:** Protein target amino acids for molecular docking.

Amino Acids Hits	Papain-Like Protease (PLpro) (6W9C)	RNA Dependent RNA Polymerase (RdRp) (6M71)	Spike Protein (S Protein) (6M17)
	ASP 286, HIS 272, and CYS 111	ARG 553, ARG 555, and LYS 545	GLY 502, TYR 489, and TYR 505

## Data Availability

Data are contained within the article.

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
