# Peer review of "Docking Analysis of Some Bioactive Compounds from Traditional Plants against SARS-CoV-2 Target Proteins"

_molecules, 2022, doi:10.3390/molecules27092662_

Round 1
Reviewer 1 Report
Although the author try to put their efforts to explore potential agents for current pandemic. But based on only docking data, it can not recommended the potential benefits of mentioned natural agents. We recommend author to do some immunofluorescence based assay if possible (Reference paper; Drugs repurposed for COVID-19 by virtual screening of
6,218 drugs and cell-based assay)
Author Response
Reviewer's comments:
Although the author try to put their efforts to explore potential agents for current pandemic. But based on only docking data, it can’t recommended the potential benefits of mentioned natural agents. We recommend author to do some immunofluorescence based assay if possible (Reference paper; Drugs repurposed for COVID-19 by virtual screening of 6,218 drugs and cell-based assay)
Thank you for valuable comments
Our current work considering 10 compounds docking to three target viral spike protein (S-protein), Papain-like protease (PLpro) and RNA dependent RNA polymerase (RdRp) each at three poses which mean we done ninety separate run in this paper for accurate results and accurate recommendations. Now we continue the in vitro cell - based assay for four compounds only chicoric acid, quinine, curcumin, and demothoxycurcumin from the tested compounds (10 compounds) and soon we will publish their results and their antiviral mechanism at Vero cells.
Reviewer 2 Report
The title seems misleading, it should indicate that it is a result from a computer program
Abstract: We introduced 10 bioactive compounds derived from traditional medicinal plants and assessed their potential for inhibiting viral proteins spike protein (S-protein), Papain-like protease (PLpro) and RNA dependent RNA polymerase (RdRp).
How the potential was evaluated???
The paper is not numerated, therefore the authors have to look for each recommendation. They are in order of appearance.
viral proteins spike protein (S-protein), -> would it be better viral spike protein?
Another viral protein, Papain-like protease (PLpro) that is vital for viral replication [8] is responsible
HCV HCoV meaning
more strategies for COVID-19 treatment?? are urgently
fera (W. somnifera) (Ashwagandha) ?? Two parentheses??
ten ligands (Figure 1) that were??? previously identified
ten ligands (I can count 8) in fig 1A. If you don’t show others that amount to 10 indicate on the legend of the figure
plants Table 4 through 90 docking run (rephrase this sentence)
kcal/mol, respectively (add comma)
Cepharanthine (CEP) is a natural alkaloi(d)
In 2021 Japanese research team reported the anti-COVID-19 activity of the combining cepharanthine/nelfinavir providing synergistic antiviral effects that CEP and nelfinavir inhibit SARS-CoV-2 entry and RNA replication, respectively [56] (rewrite this sentence)
acting mechanism to ward SARS-CoV-2, (toward)
Figure 4 legend seems to not correspond with the figure
Figure 4. Chimera visualization of 6M17 docking with Chicoric acid (1), Quinine (2), Withaferin A (3) and withanolide D (4). The yellow dots show H-bonds. (There are 10 molecular structures, no yellow dots)
activities of that would make them orally active drugs in humans (revise sentence)
It (is) well known that drug molecules
Thus, that all tested ligands have good bioavailability except chicoric acid and theaflavin as TPSA are 208.12 and 217.60 respectively. (Revise sentence)
The proposed mech- anism of SARS-CoV-2 inhibition by tested compounds (is) summarized in fig 6. (Elaborate on figure 6 please, here in the manuscript aside from the figure legend)
The conclusion indicates the simulation results as a fact. There should be a paragraph indicating the advantages and limitations of this kind of simulations. Including what additional studies should be made to verify experimentally the simulation results
Having this publication by the same authors https://ejchem.journals.ekb.eg/article_142912_a2420b3c13984e080983e6cdf9f172e1.pdf what would be the originality aside from being simulations of different components
This for curcumin compounds https://ejchem.journals.ekb.eg/article_142912_a2420b3c13984e080983e6cdf9f172e1.pdf
And I could continue
I would say there is a serious dilemma with the originality of the paper
Best regards
Reviewer 3 Report
Abd El-Aziz et al. present an interesting in silico investigation regarding the ability of 10 bioactive compounds from traditional medicines to bind and inhibit several key players from SARS-CoV-2 in order to highlight their potential use in COVID-19 treatment. However, I find that the data provided is insufficient for making the conclusions of this studies valuable to the scientific community.
Major issues:
I have serious doubts on the scientific soundness of this work, as the authors seem to confuse the disease (COVID-19) with the virus (SARS-CoV-2), page 2, line 1: “COVID-19 is positive-sense RNA (30 kb) viruses”.
I do not agree with the statement in page 2: “Currently, there is no specific anti-viral drug to treat this lethal disease. [...] In this situation, in which the preventive and therapeutic agents have not been established and recommended for administration to patients, ...”. The FDA recently approved Paxlovid (co-packaged nirmatrelvir and ritonavir) for the treatment of mild-to-moderate COVID-19 symptoms (see https://www.fda.gov/news-events/press-announcements/coronavirus-covid-19-update-fda-authorizes-first-oral-antiviral-treatment-covid-19), as well as Lagevrio (molnupiravir) for the same purpose (see https://www.fda.gov/news-events/press-announcements/coronavirus-covid-19-update-fda-authorizes-additional-oral-antiviral-treatment-covid-19-certain), while the EMA issued recommendations for the pre-market approval use of molnupiravir in treating COVID-19. Thus, Paxlovid and Lagevrio are the first two specific antiviral drugs for this disease. Please revise your introduction as to include this information, as it is recent and of interest to your manuscript.
The images generated from Chimera are absolutely terrible, I can hardly see any details. In Figure 5A, you can’t even properly see the ligand. Please find appropriate angles, color active site residues differently from other residues, set ribbon transparency and play with lighting for more proper visualization of interaction details. Please use ball-and-stick representation and a different color for your ligand as well as a different color for labels. Perhaps it is a better idea to add amino acid labels in another program. Do not generate images while ligands are selected in the software. Make H-bonds thicker. Try something at least similar to Figure 6(A) in this paper: https://journals.plos.org/plosone/article?id=10.1371/journal.pone.0161894 , which was also generated in Chimera. However, I would suggest using a white/transparent background, because details are easier to follow.
From my experience, molecular docking of potential inhibitors on the crystal structure of an unbound protein (apo form, as the authors did here) will most likely lead to faulty conclusions, as it is known that upon inhibitor binding, most proteins will undergo a conformational change in the binding site. This has also been seen recently with SARS-CoV-2’s PLPro, where the inhibitor GRL0617 bound to the active site and caused a conformation change in the BL2 loop (spanning residues 267-271) [see Gao et al. (2021) Crystal structure of SARS-CoV-2 papain-like protease. Acta Pharm Sin B. 11(1):237-245. doi: 10.1016/j.apsb.2020.08.014]. In the referenced paper, the authors suggest that PLPro adopts an induced-fit mechanism to accommodate inhibitors, highlighting the need to perform docking on more than one crystal structure of PLPro (both apo and holo forms) or perform molecular dynamics simulations of docked ligands in order to observe the stability of formed interactions upon docking. The lack of subsequent stability investigations through molecular dynamics is a downside of this manuscript. If possible, please perform molecular dynamics simulations for your docked poses or perform docking on other crystal structures of the investigated proteins as well. At the same time, 10 conformations per ligand seems rather low for flexible docking. You could try 100 poses and clustering results in Autodock to see which population is more numerous.
In their discussion, the authors highlight various literature studies which address the bioactive effects of docked ligands, but do not present any evidence into the in vitro activity of the docked ligands on the corresponding SARS-CoV-2 proteins (except in very few cases, such as cepharanthine, where some references are provided). While it is an interesting read, the presented discussion is beyond the scope of this article and should be revised as to mostly include literature data regarding activity of docked compounds against SARS-CoV-2.
It is unclear from your experimental section what the center and dimension of the gridbox used for docking was. Please also include this information, as it is essential. How did you validate your docking algorithm?
While the proposed mechanism of SARS-CoV-2 inhibitors given at the end of this manuscript is interesting, I find it purely speculative without any supporting in vitro data. In addition, this Figure is not at all discussed in the Results and Discussion section. Good docking results are not sufficient for drawing conclusions upon in vitro behavior. If possible, employ in vitro testing for all the compounds to demonstrate their presumed biological activity, such as has been done in one of their cited references for cepharanthine - Ohashi H et al. (2021) Potential anti-COVID-19 agents, cepharanthine and nelfinavir, and their usage for combination treatment. iScience. 24(4):102367. doi: 10.1016/j.isci.2021.102367.
Minor issues:
The abbreviations ‘HCoVs’ does not have an extended form, even though it is evident that it refers to human coronaviruses (page 2, line 2). Abbreviations should not be used before their extended forms are used. At the same time, please use abbreviations once they have been introduced e.g. RdRp is abbreviated at least three times (page 2, line 3 and lines 7-8, line 30).
Please be consistent in amino acid notation throughout the text. If you have started with 1-letter notation followed by number (e.g. R403 - lines 18-19), do not switch to another format e.g. 3-letter notation followed by number (e.g. Cys111 - lines 24-25 or ARG 553 - line 32). You can use any of them as long as you only use one throughout the text.
I am unsure why Table 4 is the first one in the manuscript. Please check numbering.
Figure captions should be revised, as it seems that some are not in place.
Molecular formulas in Table 2 are not all formatted properly - see compounds 8 and 9
The in silico results presented in this article is not supported by any in vitro data, but only by modeled pharmacokinetic data. For me, unsupported in silico data (especially from docking experiments) is not sufficient for publication in a reputable journal, such as Molecules. Perhaps the authors should consider a lower impact factor journal after having addressed some of the issues presented in this review.
In conclusion, this work describes the application of a well-established in silico drug design technique, with no in vitro evaluation of the compounds’ activities or stability of the generated conformations, and I recommend that the authors thoroughly review their findings before trying to publish again.
Round 2
Reviewer 1 Report
Although, author improved the manuscript a lot. However, Abstract should be rewrite as author immediately start the abstract manuscript from material and methods and its not appropriate for reader to read the article without any background or why the author started this study?
Reviewer 2 Report
The authors performed the recommended corrections to the manuscript
Although the authors are still not presenting in vitro analyses
Therefore, to be approved the authors must present a discussion of the importance of in silico studies and how there is some important correspondence between those studies and in vitro studies.
This is an example in the literature, nevertheless, there are several ones
https://www.sciencedirect.com/science/article/pii/B9780323856621000136
Best regards
Reviewer 3 Report
I appreciate the fact that the authors added ‘docking’ in the title of the manuscript to make it clear that they did not perform any functional assays. I appreciate the fact that most of my comments have been addressed properly, although I still find some issues, which will be discussed for each section of the manuscript, as follows:
Line 20: I do not consider the term ‘assay’ to be appropriate when referring to molecular docking experiments. I believe the term ‘tool’, ‘protocol’ or ‘approach’ would be more appropriate, as ‘assay’ should only be used for in vitro or in vivo techniques, not in silico, to avoid confusion.
Lines 60-62: Please check amino acid notation more carefully - the format is consistent and correct, but AG, PHEF, and GLNQ are not amino acids.
Figure 1: I would keep the same formatting for the chemical structures for consistency: e.g. theaflavin and curcumin and demothoxycurcumin are all formatted differently. Could you redraw the structures in a specialized program?
Table 1: Please use italics for all plant species
Figures 3,4,5: I appreciate that you remade the figures from Chimera, some of them are better, but they can still be improved, especially where the ligand has been cut out partially from the picture (e.g. Fig. 3(2), Fig. 3(4) and Fig. 5(1)). I insist that you find more appropriate angles, color active site residues differently from other residues, set ribbon transparency and play with lighting for more proper visualization of interaction details and regenerate these figures.
Regarding your response to my comment related to the lack of molecular dynamics, I found that the first reference you gave actually contains molecular dynamics experiments which aim to investigate the stability of the docked conformations, and actually proves my point that adding molecular dynamics experiments would be a plus to your manuscript. I feel that you have not taken my comment seriously and still urge you to do some molecular dynamics experiments to investigate the stability of docked conformations in the binding sites of the investigated proteins in order to bring more value to your manuscript. The second article is your own, but in a low impact factor journal. After reading the referenced paper, I could find very few methodological differences with the current work, and I believe that in its form, is not valuable enough to be published in a higher impact factor journal than the previously published work.
Regarding your response to my comment highlighting that very few of the docked compounds actually have been tested in vitro against SARS-CoV-2, I understand that plants containing these compounds have been used by various communities to treat the infection, but not in a controlled environment. Or are there any papers you can reference? Without any in vitro data, you cannot draw any conclusion regarding a possible mechanism if you don’t actually know if they are active! Therefore, your proposed mechanism is purely speculative. Although you have included a phrase in which you mention that further in vitro investigations should be performed, it is not sufficient to support your proposed mechanism.
I believe that you did not understand my question regarding the gridbox. When docking in Chimera, you can define the gridbox used for docking for every protein structure. What was the center and dimension of the gridbox used? What original manuscript are you mentioning?
Without any additional in vitro data or molecular dynamics experiments, I fear that this manuscript is still not valuable enough for publication in Molecules in its present form.
